# Unveiling Epigenetic Vulnerabilities in Triple-Negative Breast Cancer through 3D Organoid Drug Screening

**DOI:** 10.3390/ph17020225

**Published:** 2024-02-08

**Authors:** Xinxin Rao, Zhibin Qiao, Yang Yang, Yun Deng, Zhen Zhang, Xiaoli Yu, Xiaomao Guo

**Affiliations:** 1Department of Radiation Oncology, Fudan University Shanghai Cancer Center, Shanghai 200032, China; xxrao21@m.fudan.edu.cn (X.R.); zbqiao23@m.fudan.edu.cn (Z.Q.); 17211140003@fudan.edu.cn (Y.Y.); 15111230011@fudan.edu.cn (Y.D.); zhen_zhang@fudan.edu.cn (Z.Z.); 2Department of Oncology, Shanghai Medical College, Fudan University, Shanghai 200032, China; 3Shanghai Key Laboratory of Radiation Oncology, Shanghai 200032, China

**Keywords:** triple-negative breast cancer, patient-derived organoids, drug screening, epigenetic inhibitors, treatment

## Abstract

Triple-negative breast cancer (TNBC) poses a therapeutic challenge due to its aggressive nature and lack of targeted therapies. Epigenetic modifications contribute to TNBC tumorigenesis and drug resistance, offering potential therapeutic targets. Recent advancements in three-dimensional (3D) organoid cultures, enabling precise drug screening, hold immense promise for identifying novel compounds targeting TNBC. In this study, we established two patient-derived TNBC organoids and implemented a high-throughput drug screening system using these organoids and two TNBC cell lines. Screening a library of 169 epigenetic compounds, we found that organoid-based systems offer remarkable precision in drug response assessment compared to cell-based models. The top 30 compounds showing the highest drug sensitivity in the initial screening were further assessed in a secondary screen. Four compounds, panobinostat, pacritinib, TAK-901, and JIB-04, targeting histone deacetylase, JAK/STAT, histone demethylases, and aurora kinase pathways, respectively, exhibited potent anti-tumor activity in TNBC organoids, surpassing the effect of paclitaxel. Our study highlights the potential of these novel epigenetic drugs as effective therapeutic agents for TNBC and demonstrates the valuable role of patient-derived organoids in advancing drug discovery.

## 1. Introduction

Triple-negative breast cancer (TNBC), accounting for approximately 15–20% of all breast cancer cases, is a highly aggressive subtype with a notoriously poor prognosis [1,2]. Characterized by the absence of estrogen receptor (ER), progesterone receptor (PR), and human epidermal growth factor receptor 2 (HER2), TNBC presents a distinctive molecular profile that renders traditional hormonal and HER2-targeted therapies ineffective. Recent clinical advances have introduced targeted therapies in TNBC management, including PARP inhibitors and immunotherapy. PARP inhibitors have demonstrated particular effectiveness in BRCA-mutated cases, as evidenced by the OlympiAD [3,4] and OlympiA trials [5], which indicated that Olaparib improved survival rates in metastatic and early-stage HER2-negative breast cancer. Further, the EMBRACA and NEOTALA trials have shown the efficacy of talazoparib in advanced and early-stage breast cancer with BRCA mutations, highlighting its potential in TNBC treatment [6,7]. Additionally, recent advancements in immunotherapy, particularly immune checkpoint inhibitors targeting CTLA-4, PD-1, and PD-L1, have shown notable efficacy in enhancing TNBC treatment outcomes. Key clinical trials such as KEYNOTE-086 [8,9], IMpassion130 [10,11], and KEYNOTE-355 [12,13] have explored the therapeutic potential of PD-1 inhibitors such as pembrolizumab in metastatic TNBC, revealing significant improvements in patient prognosis. Concurrent studies, including IMpassion031 [14] and I-SPY2 [15], indicate that combining atezolizumab with chemotherapy in neoadjuvant settings significantly improves pathological complete response rates in early-stage TNBC.

Despite these advancements, anthracyclines and/or taxanes-based cytotoxic chemotherapy remain the standard treatment approach for TNBC. The high treatment resistance and increased risk of recurrence in TNBC patients [1,16,17,18,19,20] underscore the ongoing need for innovative therapeutic strategies in its management.

Epigenetic modifications are crucial in the initiation and progression of various cancers [21,22,23,24], including TNBC. The intricate interplay among DNA methylation, histone modifications, and non-coding RNA molecules contributes to the dysregulation of gene expression, culminating in the oncogenic phenotype observed in TNBC [25]. Unraveling the epigenetic vulnerabilities within TNBC holds immense promise for the exploration of innovative therapeutic strategies in TNBC management. While certain epigenetic drugs have gained regulatory approval for the treatment of hematologic malignancies, the efficacy of first-generation epigenetic drugs in solid tumor patients has been somewhat disappointing. However, with the development of novel compounds and an enhanced understanding of cancer biology, epigenetic drugs have achieved success in selected subtypes of solid tumors. For instance, clinical studies have confirmed the efficacy of HDAC inhibitors in hormone receptor-positive breast cancer [26,27]. Notably, TACH101, a novel small molecule from Tachyon, has shown promising anti-proliferative effects in diverse cancer types, including TNBC [28]. Clinical trials initiated in 2023 for advanced and metastatic solid tumors are currently evaluating its therapeutic potential. However, specific and conclusive evidence regarding the effectiveness of epigenetic drugs for TNBC remains scarce, underscoring the need for continued research and development in this field.

In recent years, three-dimensional (3D) organoid cultures have emerged as advanced models in cancer research [29,30,31,32,33]. Unlike traditional two-dimensional cell cultures, 3D organoids better recapitulate the in vivo tumor microenvironment, offering a physiologically relevant and effective platform for drug screening and discovery [34,35,36]. In contrast, patient-derived xenograft (PDX) models, while valuable, are hindered by several limitations, including low transplantation success rates, high costs, prolonged development cycles, and limited throughput in drug screening. Organoid models effectively overcome these challenges, offering advantages such as accelerated drug screening, higher throughput, enhanced clinical relevance, and improved predictive accuracy. This approach holds great promise for identifying novel compounds that specifically target TNBC cells while sparing normal tissues.

In this study, we aim to decode the epigenetic vulnerabilities inherent in TNBC using a 3D organoid drug screening approach. By combining high-throughput screening technologies with in vitro organoid phenotypic experiments, we seek to identify compounds that selectively target epigenetically dysregulated pathways in TNBC cells. Through this investigation, we anticipate uncovering novel therapeutic targets and drug candidates with the potential to effectively treat TNBC.

## 2. Results

### 2.1. Establishment of Patient-Derived TNBC Organoids and a High-Throughput Drug Screening System

Building upon the work of Hans Cleves et al. [31,37], we refined methodologies and culture conditions, successfully generating two organoids derived from TNBC patients. Patient characteristics of the corresponding organoids are summarized in Table 1. Under brightfield microscopy, the majority of the established TNBC organoid lines, TNBC1 and TNBC2, were observed as solid organoids of varying sizes (Figure 1A). Additionally, a small proportion of cystic/solid mixed-morphology organoids were identified in the TNBC2 line. To confirm their epithelial origin, we performed immunofluorescence staining for E-cadherin, cytokeratin 5, and cytokeratin 7. The TNBC organoids showed a high abundance of cytokeratin 7-positive cells, with some organoids exhibiting a peripheral distribution of cytokeratin 5-positive cells (Figure 1B). E-cadherin expression was also relatively high, indicating strong epithelial characteristics (Figure 1B). Histologically, the established two TNBC organoids closely resembled the original tumor tissue, exhibiting a high degree of consistency in protein expression profiles for ER, PR, HER2, and Ki67 (Figure 1C). These organoids exhibited stable long-term expansion capacity, making them suitable for high-throughput drug screening against TNBC.

To identify novel epigenetic compounds specifically targeting TNBC, we adapted Mao et al.’s and Chen et al.’s organoid-based drug screening system [38,39], establishing a high-throughput drug screening platform for TNBC (Figure 1D). For the preliminary screening, two TNBC cell lines (MDA-MB-231 and CAL-51) were utilized concurrently to assess the accuracy and sensitivity of the organoid-based system. All compounds were tested at a concentration of 10 μmol during the initial phase. Based on the preliminary results, the top 30 compounds displaying the most promising effects on TNBC were selected for further secondary screening. In the secondary screening, each selected compound was tested at four different concentration gradients. A comprehensive library of 169 commercially available epigenetic compounds (#L1900; Selleckchem) targeting diverse pathways, including histone deacetylase, JAK/STAT, and others, was employed for the screening process (Figure 1E, Appendix A). This selection allowed for exploration of a broad spectrum of potential epigenetic regulators.

### 2.2. Growth Inhibition Profiling of Epigenetic Drugs on TNBC Organoids and Cell Lines

The preliminary screening in our investigation provided intriguing insights into the growth inhibition profiles of epigenetic drugs on TNBC organoids and cell lines. Organoid-based systems offer remarkable precision in drug response assessment compared to cell-based models, revealing nuanced efficacy gradients among drugs (Figure 2). Notably, TNBC organoids maintained the inherent heterogeneity in drug sensitivity observed in the original tumors. Organoids derived from different patients displayed disparities in response to specific drugs. For instance, JAK/STAT inhibitors (baricitinib, ruxolitinib, Tofacitinib, Oclacitinib), HDAC inhibitors (tubastatin A HCl, MC1568, mocetinostat), and aurora kinase inhibitors (barasertib, danusertib) elicited differing responses between TNBC1 and TNBC2 organoids, with TNBC1 organoids showing overall heightened sensitivity (Figure 2 and Appendix A). In contrast, the two cell lines exhibited a relatively uniform response across various drugs.

Interestingly, certain drugs elicited similar responses in both TNBC organoids and cell lines. HDAC inhibitors (CUDC-907, LAQ824, panobinostat), aurora kinase inhibitor (TAK-901), and JAK/STAT inhibitors (AG-490, pacritinib) demonstrated potent inhibitory effects in both models. Conversely, drugs with limited efficacy in TNBC organoids, such as EPZ004777 and Remodelin, also showed minimal growth inhibition in TNBC cell lines (Figure 2). This parallelism suggests common epigenetic or pharmacological mechanisms at play in TNBC, providing crucial insights for the development of targeted therapies.

To identify potent compounds against TNBC, we systematically organized preliminary drug screening results based on the average drug sensitivity (expressed as the reciprocal of the growth inhibition rate) across two TNBC organoids and two cell lines for each compound (Appendix A). The top 30 drugs with the highest average sensitivity were then selected for secondary screening.

### 2.3. Dose–Response Characterization and Drug Sensitivity Analysis of Selected Drugs in TNBC Organoids

To elucidate the specific anticancer properties of the 30 drugs identified in the preliminary screening, a secondary screening was conducted on two TNBC organoids. Each drug was assessed across four concentration gradients. The selected drugs included 13 HDAC inhibitors, five JAK/STAT inhibitors, two aurora kinase inhibitors, two histone demethylase inhibitors, two histone methyltransferase inhibitors, two DNA methyltransferase inhibitors, one Pim inhibitor, one DNA/RNA synthesis inhibitor, and one topoisomerase inhibitor. Consistent with the initial screening results, all the selected compounds demonstrated growth inhibitory effects on the TNBC organoids (Figure 3A,B). Notably, specific compounds such as certain HDAC inhibitors (e.g., panobinostat and CUDC-907), the histone demethylase inhibitor JIB-04, the JAK/STAT pathway inhibitor pacritinib, and the aurora kinase inhibitor TAK-901 significantly reduced cell viability in both organoids. This indicates a potential therapeutic window for these agents in TNBC treatment. Moreover, variations in drug responses between the two organoids were observed, suggesting patient-specific sensitivities or resistances. TNBC1 organoids showed an overall heightened sensitivity to epigenetic drugs compared to TNBC2 organoids. These findings further validate the robustness and reliability of our high-throughput drug screening system.

To quantify the tumor-killing effect of these drugs on TNBC organoids, GR_50_ values for each compound were calculated (Figure 3B). The most potent compounds among the epigenetic inhibitors targeting the HDAC, JAK/STAT, histone demethylases, and aurora kinase pathways—namely panobinostat, pacritinib, TAK-901, and JIB-04—demonstrated strong efficacy against both TNBC organoids. Our results highlight the potential therapeutic role of these agents in TNBC treatment, though further investigation is required to confirm their actual therapeutic efficacy in TNBC.

### 2.4. Superior Cytotoxicity of Novel Agents Panobinostat, Pacritinib, TAK-901, and JIB-04 over Paclitaxel in TNBC Organoid Models

To visually ascertain the cytotoxic effects of the selected compounds—panobinostat, pacritinib, TAK-901, and JIB-04—on TNBC organoids, we employed the established Calcein-AM/PI staining and imaging methodology [40]. This technique differentiates between viable (green fluorescence) and dead (red fluorescence) cells, providing a clear indication of cytotoxicity. The compounds were administered two days post-passage of the organoids, and their effects were evaluated after five days. DMSO was used as the negative control, and paclitaxel, a widely used chemotherapeutic agent, served as a comparative control.

As demonstrated in Figure 4A–C, treatment with paclitaxel resulted in reduced organoid volumes compared to the DMSO control group (TNBC1: 10.20 ± 3.99 × 10^4^ pixels vs. control 36.99 ± 8.23 × 10^4^ pixels, *p* < 0.05; TNBC2: 17.62 ± 4.56 × 10^4^ pixels vs. control 43.05 ± 13.12 × 10^4^ pixels, *p* = 0.0798). However, the paclitaxel-treated organoids maintained their structural integrity and contained a significant proportion of viable cells (TNBC1: 50.83 ± 12.10%; TNBC2: 72.23 ± 9.38%). This observation suggests that the therapeutic effect of paclitaxel on TNBC is limited, primarily inhibiting the growth of TNBC organoids. In contrast, organoids treated with panobinostat, pacritinib, TAK-901, and JIB-04 showed almost complete disintegration of basic morphology, indicating significant cytotoxicity (viable cell area for TNBC1: 0.56 ± 0.15 × 10^4^ pixels, 1.30 ± 0.52 × 10^4^ pixels, 0.89 ± 0.33 × 10^4^ pixels, 0.69 ± 0.18 × 10^4^ pixels, respectively, vs. control 36.99 ± 8.23 × 10^4^ pixels, all *p* < 0.001; TNBC2: 1.16 ± 0.56 × 10^4^ pixels, 2.05 ± 1.28 × 10^4^ pixels, 1.31 ± 0.28 × 10^4^ pixels, 1.12 ± 0.40 × 10^4^ pixels, respectively, vs. control 43.05 ± 13.12 × 10^4^ pixels, all *p* < 0.001). These organoids predominantly consisted of dead cells with very few live cells remaining (viable cell percentage: TNBC1: 18.33 ± 8.50%, 27.74 ± 10.91%, 23.22 ± 9.64%, 17.94 ± 9.75% vs. control 97.38 ± 5.86%, all *p* < 0.001; TNBC2: 24.67 ± 9.62%, 26.94 ± 11.94%, 23.18 ± 12.13%, 18.84 ± 9.23% vs. control 95.64 ± 2.39%, all *p* < 0.001). These findings collectively underscore the tumor-killing effects of panobinostat, pacritinib, TAK-901, and JIB-04 on TNBC organoids, significantly surpassing that of the standard chemotherapeutic agent paclitaxel. To evaluate the selectivity of these compounds towards cancer cells, we further compared their cytotoxicity against non-neoplastic cells, using the MCF-10a cell line. The results reveal that panobinostat, pacritinib, TAK-901, and JIB-04 exhibit lower cytotoxicity towards non-neoplastic cells compared to paclitaxel (Appendix A). This highlights the potential of these novel compounds as more effective therapeutic agents against TNBC, warranting further in vivo investigation.

## 3. Discussion

In this study, we developed a platform for drug screening using patient-derived TNBC organoids, which identified novel epigenetic compounds with potent anti-tumor activity. Among the identified compounds, panobinostat targeting histone deacetylase, pacritinib targeting JAK/STAT, JIB-04 targeting histone demethylases, and TAK-901 targeting aurora kinase exhibited superior efficacy in suppressing TNBC growth compared to paclitaxel. Our study demonstrates the potential of PDOs in drug discovery for TNBC. The identification of novel epigenetic compounds with potent anti-tumor effects represents a significant step forward in the development of new therapies for TNBC.

Building upon the foundational work of Hans Cleves and his team in establishing a culture system for breast cancer PDOs [31], subsequent research has expanded its applications. Ping Chen and colleagues adapted this system for organoid drug screening, aiding personalized treatments for advanced breast cancer patients [39]. Sonam et al. established a TNBC-focused PDO biobank, using single-cell sequencing to show the potential of TNBC PDOs in elucidating TNBC biology and progression [41]. Kelvin et al. leveraged this platform to identify targets such as NCOR2/HDAC3 for overcoming treatment resistance, enhancing patient outcomes [42]. Our research further extends Cleves’ approach, implementing high-throughput screening of epigenetic drugs to decode epigenetic vulnerabilities in TNBC. In line with these studies, our established TNBC PDOs, which accurately reflect the original tumor’s histology and protein expression, provide a robust model for drug sensitivity assessment. Our drug screening system, inspired by the work of Mao et al. [38] and Chen et al. [39], enables rapid evaluation of epigenetic drug efficacy within a week, accelerating the discovery and optimization of new treatments for TNBC.

The pioneering work of Vlachogiannis G’s team demonstrated the effectiveness of PDOs in guiding clinical treatment for patients with metastatic gastrointestinal cancer, highlighting the high accuracy of organoid drug screening with 93% specificity, 100% sensitivity, 88% positive predictive value, and 100% negative predictive value [43]. Our research aligns with these findings, showing that organoid-based systems possess a remarkable ability to closely mimic the in vivo drug response compared to cell-based models, effectively capturing nuanced drug efficacy gradients. Importantly, our TNBC organoids preserved the original tumor’s inherent heterogeneity in drug sensitivity, in contrast to the almost consistent responses observed in cell lines. This underscores PDOs’ ability to serve as a more reliable platform for predicting actual clinical responses.

While organoid-based high-throughput drug screening more accurately mimics clinical responses than cell-line-based methods, our study found similar responses between TNBC organoids and cell lines to certain compounds. This indicates shared epigenetic regulatory or drug reaction mechanisms in TNBC patients, providing crucial insights for the development of targeted TNBC treatments. Following this, we conducted a secondary screening of the top 30 compounds most effective against both organoids and cell lines. The results from this secondary screening corroborated our initial findings, demonstrating notable growth inhibitory effects across various drug classes. The compounds targeting histone deacetylase, JAK/STAT, histone demethylases, and aurora kinase pathways—specifically panobinostat, pacritinib, TAK-901, and JIB-04—showed significant tumor-killing effects on TNBC. These effects were further validated through Calcein-AM/PI staining and imaging, confirming the high potential of these drugs as effective agents against TNBC.

Extensive research has been conducted on the efficacy of HDAC inhibitors in the treatment of TNBC [44,45]. Chandra et al.’s research highlights the potential of panobinostat in targeting aggressive TNBC cells [46]. Nicoletta et al. found that panobinostat enhances E-cadherin expression in TNBC cells, reducing their invasiveness [47]. Aparna’s research implicated HDACs 1, 2, and 3 in vasculogenic mimicry in TNBC, highlighting the therapeutic importance of HDACs [48]. Beyond HDACs, the JAK/STAT pathway is also implicated in TNBC development and drug resistance [49,50,51], with studies showing that combining JAK inhibitors with SMO-GLI1/tGLI1 inhibitors significantly reduces TNBC progression [52]. The potential of the histone demethylase inhibitor JIB-04 to enhance the treatment sensitivity of resistant triple-negative inflammatory breast cancer cells has also been explored [53]. Furthermore, the critical role of aurora kinase in TNBC progression and chemoresistance has been well-established [54,55,56]. Aurora kinase inhibitors have been shown to effectively inhibit TNBC cells, particularly when used in combination with other drugs [56,57]. A phase II clinical trial also demonstrated that the aurora kinase inhibitor ENMD-2076 offers clinical benefits for certain TNBC patients [58]. These research findings are consistent with the results from our study using patient-derived TNBC organoids for drug screening. Our study is the first to demonstrate the efficacy of compounds targeting histone deacetylase, JAK/STAT, histone demethylases, and aurora kinase pathways—specifically panobinostat, pacritinib, TAK-901, and JIB-04—in killing TNBC cells using patient-derived organoids. This highlights the potential of our organoid-based approach for identifying promising therapeutic options and contributes to the development of new treatment strategies for TNBC.

However, it is important to note that our results, while promising, should be considered as foundational for further research rather than definitive evidence of clinical efficacy. The in vitro nature of organoid models may not fully capture the complexity of tumor biology in vivo. The translation of these findings to clinical practice requires careful consideration of pharmacodynamics, pharmacokinetics, and potential side effects in humans. Furthermore, our study involved only two TNBC PDOs and two TNBC cell lines, and, given the heterogeneity among TNBC patients, further research is needed to identify the best-suited patient population for these epigenetic drugs.

## 4. Materials and Methods

### 4.1. Sample Collection

Tumor tissues were collected from breast cancer patients after obtaining informed consent. The collection was ethically approved by the Fudan University Shanghai Cancer Center’s Review Board (protocol 050432-4-2108, 9 August 2021). Written consents were also secured for publication purposes.

### 4.2. Tissue Processing and Organoid Culture

Our methodology for organoid derivation and culture was adapted from established protocols [31,37] with minor modifications. Specifically, for organoid derivation, the tissue was finely minced (<1 mm^3^) until it appeared uniform and somewhat viscous. We modified the enzymatic digestion process by utilizing 1.5 mg/mL collagenase type III (#LS004182; Worthington Biochemical Corporation, Worthington, OH, USA) instead of Collagenase Type II. Collagenase type III, known for its lower protease activity, facilitates the digestion of breast cancer tissues into single cells while preserving cell viability. The detailed procedure is as follows: Upon receipt, tissues were minced with a sterile surgical blade and enzymatically digested into cell aggregates using 1.5 mg/mL collagenase type III (#LS004182; Worthington) at 37 °C for 1–2 h. The resulting cell aggregates were filtered through a 100 µm strainer and washed twice with PBS++ (cold PBS containing 1% BSA and 1× P/S). After centrifugation at 200× *g* for 5 min, the pellet was resuspended in Matrigel (#356231; Corning, Somerville, MA, USA) and seeded onto a pre-warmed 24-well plate. The Matrigel-embedded organoids were then solidified at 37 °C for 10 min. Following solidification, 1 mL of breast cancer organoid medium (specific formulation described below) was added to each well, with the plate incubated at 37 °C. The medium was refreshed every 3–4 days.

The composition of our culture medium underwent slight modifications compared to the previously established protocol [31,37]. The earlier protocol recommended using R-spondin-1-conditioned medium and Noggin-conditioned medium, but, due to their unavailability and instability in our laboratory, we opted for commercial alternatives: 250 ng/mL Rspo1 (#11083-HNAS; Sino Biological Inc., Beijing, China) and 100 ng/mL Noggin (#50688-M02H; Sino Biological Inc.). Additionally, we adjusted the concentration of SB 202190 to 500 nM (#S7067, Sigma, St. Louis, MO, USA), as opposed to the 1 µM used previously, based on preliminary tests indicating more effective growth acceleration of breast cancer organoids at this concentration. The breast cancer organoid medium was prepared by supplementing Advanced DMEM/F12 (#12634-028; Gibco, Waltham, MA, USA) with the following components: 1× GlutaMAX (#35050061; Gibco), 1× P/S (#10378-016; Gibco), 1× HEPES (#15630080; Gibco), 1× B27 (#17504-044; Invitrogen, Waltham, MA, USA), 1× Promicin (#ant-pm-2; InvivoGen, San Diego, CA, USA), 1.25 μM N-acetylcysteine (#A9165; Sigma), 10 mM Nicotinamide (#N0636; Sigma), 250 ng/mL Rspo1 (#11083-HNAS; Sino Biological Inc.), 100 ng/mL Noggin (#50688-M02H; Sino Biological Inc.), 5 ng/mL EGF (#AF-100-15; PeproTech, Cranbury, NJ, USA), 0.5 μM A83-01 (#2939; Tocris, Bristol, UK), 500 nM SB202190 (#S7067, Sigma), 5 nM Neuregulin 1 (#100-03; PeproTech), 5 ng/mL FGF7 (#100-19; PeproTech), 20 ng/mL FGF10 (#100-26; PeproTech), and 5 μM Y27632 (#S1049, Selleck, Tokyo, Japan).

### 4.3. Organoid Passaging

Breast cancer organoids underwent passaging every 3–6 weeks at a ratio of 1:3–1:5. Briefly, organoids were washed twice with cold PBS++, followed by centrifugation. The organoids were then suspended in 2–4 mL TrypLE Express (#12605028; Gibco), and the suspension was incubated at 37 °C for 20 min to facilitate enzymatic digestion. Regular microscopic monitoring ensured proper digestion. Subsequently, the cells underwent two additional washes with cold PBS++ and centrifugation at 200× *g* for 5 min. The resulting cell pellet was resuspended in Matrigel, reseeded in 24-well plate wells, and cultured as previously described.

### 4.4. Cell Cultures

The human TNBC cell line MDA-MB-231 and human normal breast epithelial cell line MCF-10a were procured from ATCC, while the TNBC cell line CAL-51 was obtained from DSMZ. MDA-MB-231 and CAL-51 cells were cultured in DMEM supplemented with 1% P/S and 10% fetal bovine serum (FBS). MCF-10a cells were cultured in DMEM supplemented with 10% FBS, 20 ng/mL EGF, 0.5 μg/mL hydrocortisone, 10 μg/mL insulin, 1% NEAA, and 1% P/S. The culture conditions were maintained at a constant temperature of 37 °C in a humidified environment containing 5% CO_2_.

### 4.5. Drug Screening

The screening of epigenetic compounds was conducted using the epigenetic compound library (#L1900; Selleckchem, Houston, TX, USA), consisting of 169 compounds (Appendix A). The compounds were stored at a concentration of 10 mM and maintained at −80 °C. Prior to initiating drug screening, a 10-fold working solution was prepared by individually diluting each compound in the culture medium. In each well of the screening plate, which contained 90 μL of the cell–Matrigel mixture, 10 μL of the diluted drug was added, resulting in a final drug concentration of 10 μmol/L for the primary screening phase. For the secondary screening, we applied a four-point dose dilution series spanning concentrations of 10^−7^, 10^−6^, 10^−5^, and 10^−4^ mol/L. For each drug treatment, experiments were conducted in triplicate.

For organoid-based drug screening, harvested organoids were enzymatically digested into single cells, followed by suspension in a culture medium containing 5% Matrigel (5000 cells/mL). Subsequently, the cell–Matrigel mixture was seeded into each well of a 96-well plate with a 90 μL volume per well. For cell-based drug screening, the cell suspension concentration was adjusted to 1000 cells/mL to prevent rapid overgrowth in the control group (due to cells growing as adherent monolayers). Similarly, 90 μL of the cell suspension was added to each well within a 96-well plate. To mitigate evaporation effects, ddH_2_O was added into the surrounding wells. Following a 2-day incubation period, allowing for the development of 3D organoid structures or ensuring complete cell adherence, 10 μL of the diluted drug was added to each well. DMSO served as the negative control. After a five-day incubation period, organoid viability was assessed through ATP values, using the CellTiter-Glo^®^ 3D assay (#G9683; Promega, Madison, WI, USA). The primary screening outcomes, expressed as a percentage of cell viability normalized against the control group, were graphically depicted using a heatmap generated in the R programming environment, utilizing both the ‘gplots’ and Complexheatmap packages. Subsequently, 30 drugs exhibiting the most significant tumor-killing effects were selected for a secondary screening, wherein concentration gradients were implemented. ATP values at the start and end of the drug treatment were recorded to compute Growth Rate Inhibition (GR) values. The GR_50_, representing the drug concentration at which GR attains 0.5, was computed using the R package GR metrics.

### 4.6. Histology, Immunohistochemistry and Immunofluorescence Staining

Tissues and organoids were fixed in 4% paraformaldehyde for 12–24 h and 30 min, respectively. After paraffin embedding and 4 μm sectioning, standard hematoxylin and eosin staining, immunohistochemistry, and immunofluorescence staining techniques were employed. The primary antibodies used for immunohistochemistry included anti-ER (#ab16660; Abcam, Cambridge, UK; 1:200), anti-PR (#ab101688; Abcam; 1:400), anti-HER2 (#2165; Cell Signaling Technology, Danvers, MA, USA; 1:200), and anti-Ki67 (#550609; BD Bioscience, Franklin Lakes, NJ, USA; 1:200). Immunohistochemistry utilized biotinylated secondary antibodies and the Pierce DAB substrate kit (#34002; Thermo Fischer, Waltham, MA, USA). The primary antibodies used for immunofluorescence staining included anti-cytokeratin 5 (#A11396; Abclonal, Wuhan, China; 1:100), anti-cytokeratin 7 (#ab181598; Abcam; 1:1000), and anti-E-cadherin (#14472; Cell Signaling Technology, Danvers, MA, USA; 1:200). Nuclei were counterstained with DAPI (#D571; Invitrogen, Waltham, MA, USA; 1:1000).

### 4.7. Fluorescence (Calcein-AM/PI) Staining of Organoids

Calcein-AM/PI staining of organoids was performed as previously described [40]. A 50 μg/mL stock solution of Calcein-AM (#40719ES50, Yeasen, Shanghai, China) was prepared by dissolving the powder in DMSO and stored at −20 °C. Before staining, the stock solution was diluted 1:1000 in PBS. A 10-fold diluted working solution of PI (#40755ES64, Yeasen) was prepared using PBS. Organoids were gently washed twice with PBS to remove any residual medium or debris and subsequently incubated in the prepared working solution at 37 °C. Observations were conducted subsequent to a final wash with PBS.

### 4.8. Imaging of Organoids

Post a 5-day drug treatment, organoids were checked using an Olympus IX83 Inverted Microscope, with image acquisition facilitated by Zero Drift Compensation (ZDC) technology (Olympus, Shinjuku, Tokyo, Japan).

### 4.9. Statistical Analysis

Dose–response curves were generated and statistically analyzed using GraphPad Prism software (version 9.0). Fluorescence images were processed using ImageJ (V7.0) software. Digital image data were assessed in a blinded manner with respect to the treatment conditions. To identify significant differences between experimental groups, a two-tailed Student’s t-test was used, and representative values of standard deviation (SD) or standard error of the mean (SEM) were reported. A *p*-value < 0.05 was considered statistically significant.

## 5. Conclusions

In conclusion, our study successfully established patient-derived TNBC organoids, providing an effective model for high-throughput drug screening. Utilizing this platform, we identified novel epigenetic compounds targeting histone deacetylase, JAK/STAT, histone demethylases, and aurora kinase pathways, which exhibit significant tumor-killing effects on TNBC. Among these compounds, panobinostat, pacritinib, TAK-901, and JIB-04 showed superior efficacy compared to the chemotherapeutic agent paclitaxel. Our findings highlight the potential of these compounds as promising therapeutic agents for TNBC and reinforce the value of patient-derived organoids in advancing drug discovery. Future investigations are warranted to confirm their therapeutic efficacy in clinical settings.

## Figures and Tables

**Figure 1 pharmaceuticals-17-00225-f001:**
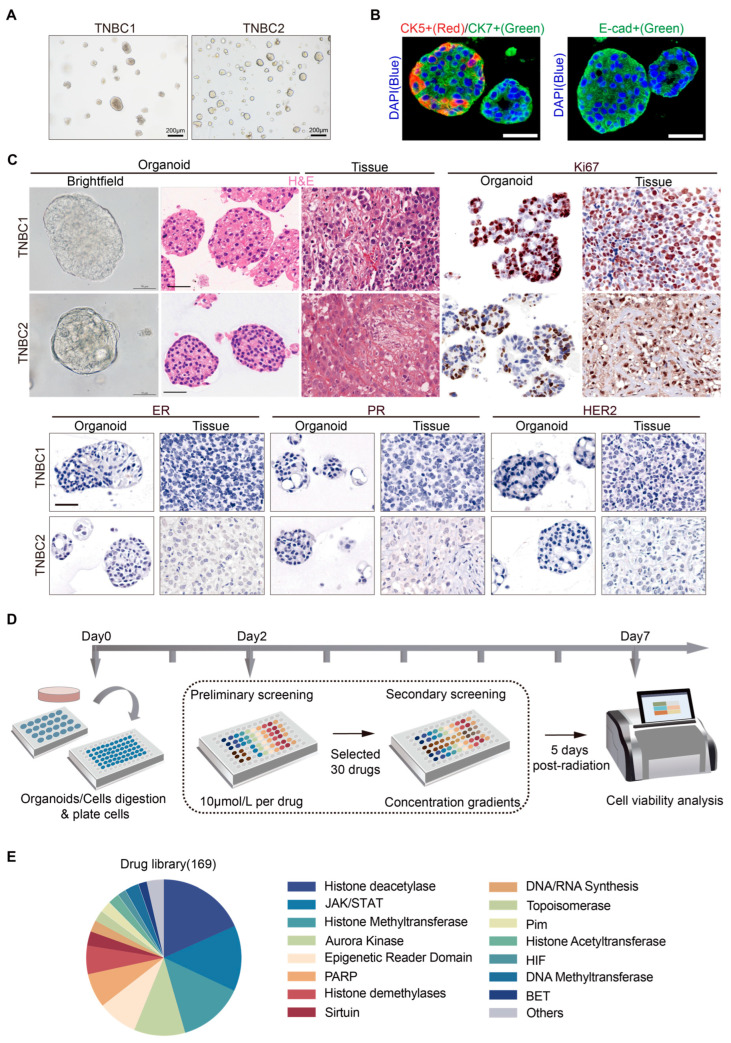
Establishment of patient-derived TNBC organoids and high-throughput drug screening system. (**A**) Representative brightfield microscopy images of two TNBC organoids; scale bars, 200 µm. (**B**) Immunofluorescence staining for cytokeratin 5 (CK5, red), cytokeratin 7 (CK7, green), and E-cadherin (E-cad, green) in TNBC organoids, with nuclear counterstaining (DAPI, blue); scale bars, 20 µm. (**C**) Representative brightfield microscopy images, H&E staining, and immunohistochemistry for Ki67, ER, PR, and HER2 of two TNBC organoids along with corresponding primary tumors; scale bars, 50 µm. (**D**) Schematic diagram illustrating the experimental workflow of epigenetic drug screening. (**E**) Overview summarizing the classification of drugs within the epigenetic drug library utilized in this study.

**Figure 2 pharmaceuticals-17-00225-f002:**
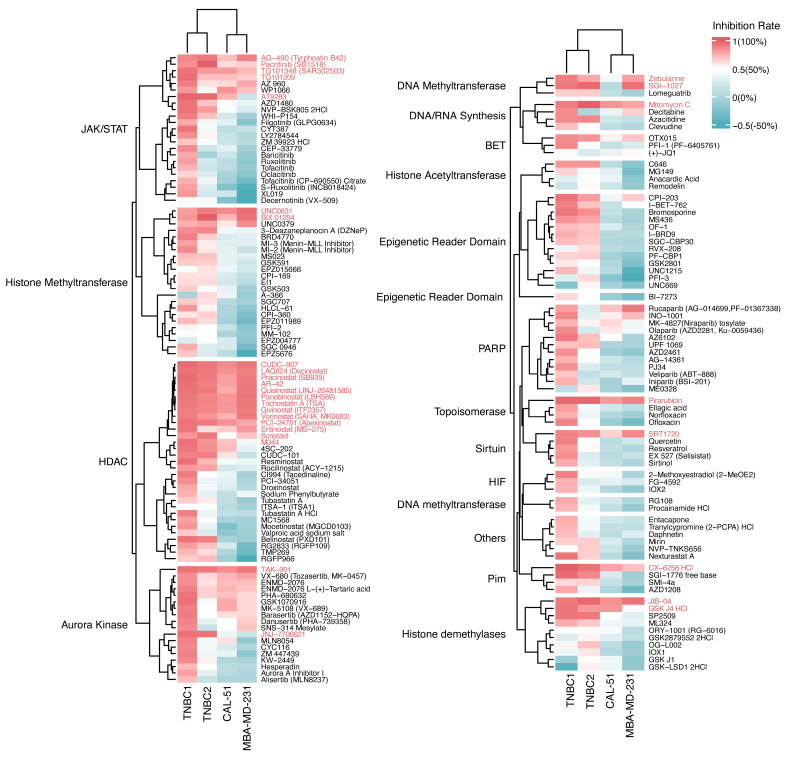
Growth inhibition profiling of 169 tested drugs on TNBC organoids and cell lines. Heatmap illustrating the growth inhibition rates of 169 tested drugs on TNBC organoids and cell lines. Molecular targets categorizing the drugs are presented on the left side of the heatmap, with specific drug names annotated on the right side. Drugs with growth inhibition rates exceeding 50% (compared to DMSO–treated groups) are colored red, signifying pronounced tumor-killing efficacy against triple-negative breast cancer, while those with rates below 50% are colored blue. Drugs selected for secondary screening are labeled in red.

**Figure 3 pharmaceuticals-17-00225-f003:**
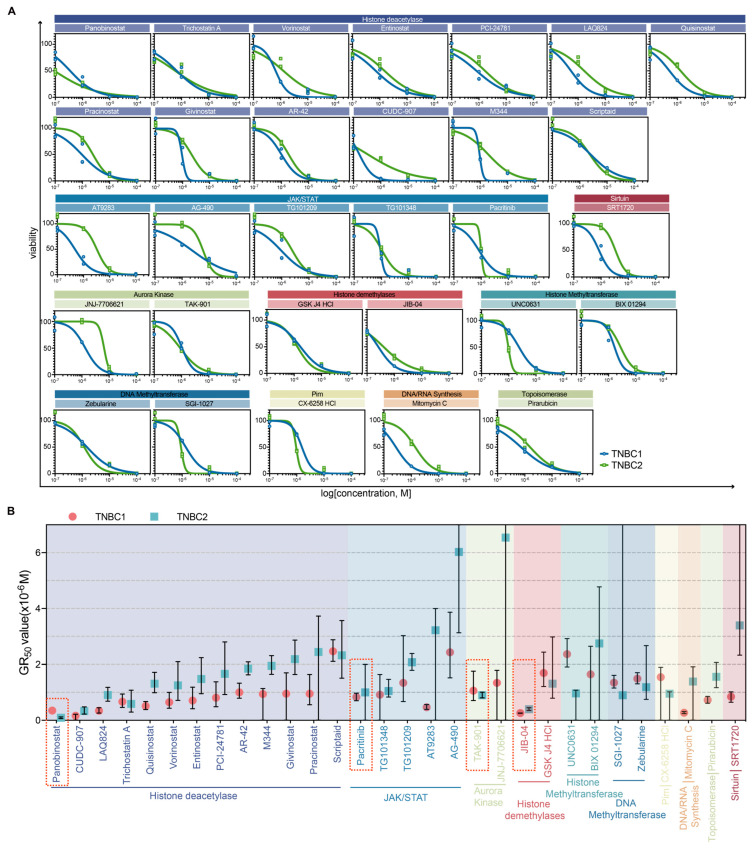
Dose–response characterization and drug sensitivity analysis of selected drugs in TNBC organoids. (**A**) Dose–response curves depicting the pharmacological effects of 30 selected drug compounds on two TNBC organoids. Each data point represents the mean value derived from three independent replicates. (**B**) The GR_50_ values for the 30 drug candidates administered to the two TNBC organoids. Drugs targeting the same molecular pathways are grouped together and distinguished by different colors. Within each targeted pathway, different drugs are arranged in ascending order based on their average GR_50_ values. The error bars represent the 95% confidence interval. The red dotted box highlights specific compounds that underwent further organoid phenotypic experiments to validate their effects on TNBC.

**Figure 4 pharmaceuticals-17-00225-f004:**
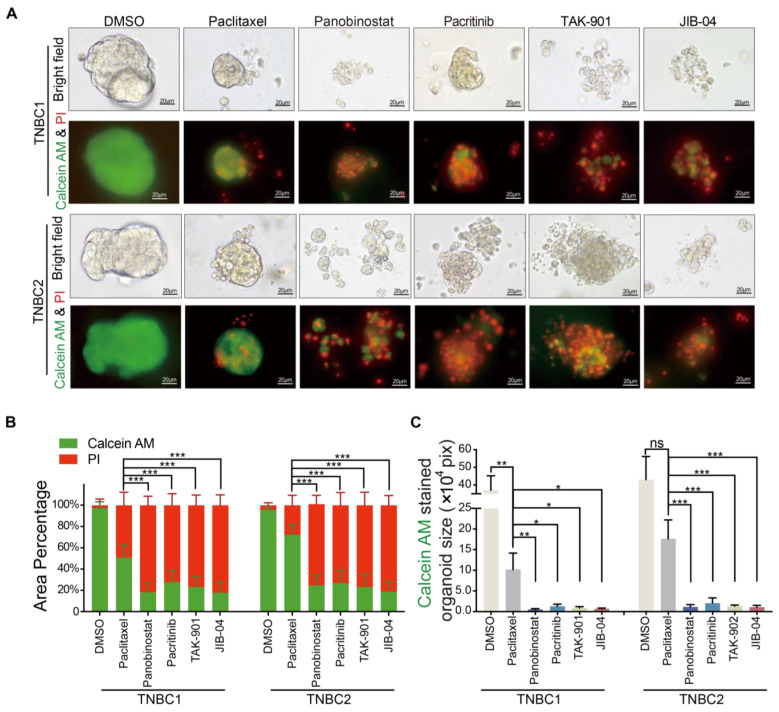
Evaluation of drug responses in TNBC organoids through imaging and quantitative analysis. (**A**) Bright-field images and corresponding fluorescence images stained with Calcein-AM and PI, depicting the response of two triple-negative breast cancer (TNBC) organoids following a 5-day exposure to 10 µM paclitaxel (serving as a chemotherapeutic drug control), panobinostat, pacritinib, TAK-901, JIB-04, and DMSO (utilized as a negative control). Calcein-AM and PI effectively distinguish living and deceased cells within patient-derived TNBC organoids; scale bars, 20 µm. (**B**) Quantification of the percentages of Calcein-AM– or PI–stained areas in organoids after a 5-day exposure to 10 µM paclitaxel, panobinostat, pacritinib, TAK-901, JIB-04, and DMSO. (**C**) Quantification of the Calcein-AM stained areas in organoids following a 5-day exposure to 10 µM paclitaxel, panobinostat, pacritinib, TAK-901, JIB-04, and DMSO. * *p* < 0.05, ** *p* < 0.01, *** *p* < 0.001.

**Table 1 pharmaceuticals-17-00225-t001:** Patient clinical characteristics corresponding to patient-derived organoids in this study.

Patient	Age (Years)	Gender	Clinical Stage	Histological Type	Histologic Grade	Molecular Subtype	Laterality	Tumor Location	Sample Type	Pre-Sampling Therapeutic History
TNBC1	45	Female	cT2N0M0	Invasive Ductal Carcinoma	II	TNBC	Left	Central/ Medial	Surgical	Modified Radical Mastectomy
TNBC2	50	Female	cT4N1M0	Invasive Ductal Carcinoma	II	TNBC	Left	Central/ Medial	Biopsy	Neoadjuvant Chemotherapy(Paclitaxel and Carboplatin, PCb Protocol)

## Data Availability

Data are reported in the article and Appendix A.

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
