# Peer review of "Unveiling Epigenetic Vulnerabilities in Triple-Negative Breast Cancer through 3D Organoid Drug Screening"

_pharmaceuticals, 2024, doi:10.3390/ph17020225_

Round 1
Reviewer 1 Report
Comments and Suggestions for Authors
This paper focuses on a hot topic in cancer therapeutics - epigenetic therapies. In fact, TNBC lacks effective targeted therapeutic options and the approach described in this paper can contribute to find better therapeutic solution for this aggressive cancer type.
Introduction
- Would prefer that authors substitute “formidable”, as the subject is an aggressive disease leading to many death per year.
- Feel that authors do not approach 2 existing targeted therapies for TNBC: parp inhibitors for BRCA mutations and Immunotherapy; this are available therapies in the clinics, although only a reduced % of patients shows responses, they should not be ignored in the introduction. Also little amount of literature is referenced for TNBC.
- Line 49: there is a small molecule, TACH101 from Tachyon, that entered clinical trials in 2023 for solid tumours; may include this reference since it is the first epigenetic molecule to enter this phase for solid tumors.
Results
- Fig1B: 5h day? From the methods I depict that treatment duration was 5 days, correct this in the figure.
- Lines 108-110: “Similar 108 response patterns were….” This sentence does not reflect what I observe in the Figure 2; cell lines show similar pattern, but completely different inhibition profile from TNBC organoids, specially considering TNBC1. This is justified in the following text, so would suggest to rewrite this sentence for accuracy.
- Authors claim that panobinostat, pacritinib, TAK-901, and JIB-04 show higher effect than paclitaxel over TNBC organoids and can therefore be considered more effective therapeutic agents against TNBC (line 192). Would request further tests of these drugs using non-neoplastic cells to infer about selectivity of these drugs toward cancer cells. A more effective cell growth inhibition profile does not mean that the drugs are a better therapeutic option, if selectivity is not increased.
Materials and Methods:
4.1 - Where the samples come from and respective ethical issues should be available here.
4.4. - missing reference to the purity of the compounds, which should be higher than 95% for biological testing.
Formatting:
Table 1 - maybe join columns to avoid the the separation of a word in many lines
Reviewer 2 Report
Comments and Suggestions for Authors
I am writing to provide my review of the manuscript entitled "Unveiling Epigenetic Vulnerabilities in Triple-Negative Breast Cancer through 3D Organoid Drug Screening" submitted to Pharmaceuticals.
Firstly, I would like to commend the authors for their significant contribution to the field of triple-negative breast cancer (TNBC) research. The study is well-conceived, addressing the urgent need for effective therapeutic strategies against TNBC—a subtype of breast cancer notorious for its aggressive behavior and limited treatment options. The use of patient-derived TNBC organoids and the implementation of a high-throughput drug screening system represent a commendable advance in the field, offering a more accurate and relevant model for drug discovery.
The manuscript clearly outlines the potential of epigenetic modifications as therapeutic targets and successfully identifies four compounds with potent anti-tumor activity in TNBC organoids. The detailed exploration of these compounds' effects on different signaling pathways is both thorough and insightful. The study's results are promising, suggesting that panobinostat, pacritinib, TAK-901, and JIB-04 could lead to effective new treatments for TNBC patients.
However, I recommend that for the manuscript to effectively communicate its findings and implications, the authors should provide a clearer and more detailed conclusion section after the discussion. It would be beneficial for the readers to have a summarized and explicit statement of the study's main findings, the potential impact of these findings on TNBC treatment, and future directions for research. This would not only reinforce the significance of the work but also provide a concise reference point for readers interested in the key outcomes and implications of the study.
I recommend publication following minor revisions.
Thank you for the opportunity to review this manuscript.
Reviewer 3 Report
Comments and Suggestions for Authors
The authors, Rao X et al., of the submitted manuscript “Unveiling Epigenetic Vulnerabilities in Triple-Negative Breast Cancer through 3D Organoid Drug Screening” have been presented a crucial study in the field of cancer research, particularly for triple-negative breast cancer (TNBC), as it addresses the therapeutic challenges posed by the aggressive nature of TNBC and the lack of targeted treatments. The significance lies in the exploration of epigenetic modifications as contributors to TNBC development and drug resistance, offering potential therapeutic targets. Moreover, the use of patient-derived TNBC organoids in three-dimensional (3D) cultures represents a cutting-edge approach for precise drug screening, showcasing superior sensitivity compared to traditional cell-based models. The identification of top compounds, such as panobinostat and pacritinib, with potent anti-tumor activity underscores the promise of novel epigenetic drugs in TNBC therapy. Overall, the study highlights the valuable role of patient-derived organoids in advancing drug discovery for TNBC.
Although it is a very well structured, developed work, with well-executed and complex techniques, there are some issues that must be addressed for its publication:
1. The authors should explain in more detail the modifications they have made to the published protocols for obtaining and culturing organoids. These are complex techniques that differ between organoids of different origin, so in order to ensure reproduction it is necessary to provide all the data.
2. The authors have characterized the organoids for their TNBC origin, but have not done adequate characterization of the organoids, such as microscopic photography to examine their morphology (solid, cystic or grape-like morphology) and labeling to demonstrate the presence of epithelial organoid markers (such as e-cadherin and cytokeratins).
3. I think the authors erroneously state that "organoid-based systems exhibit superior sensitivity compared to cell-based models". First of all, a 3D system is not more sensitive to treatment than a 2D system because in the 3D system there is a greater cell-cell interaction that favors greater resistance and also cell-matrix interaction that hinders drug penetration. This is why a 2D monolayer system is more sensitive, as it does not have this complex microenvironment and more direct access of the drug to the cells. In relation to this, only Figure S1 shows this comparison and I consider that it should be better explained. What is the basis of this "sensitivity"? The authors indicate that "expressed as a percentage of cell viability normalized against the control group", so I imagine that the higher fluorescence of the CellTiter-Glo® 3D compound is greater cell viability and this would correspond to the organoid values being higher than those of the cells. Based on this, it is not a higher sensitivity, it would be lower sensitivity and the monolayer would be more sensitive, resulting in greater cell death.
Round 2
Reviewer 1 Report
Comments and Suggestions for Authors
The authors have revised the document in accordance with all the points that I previously mentioned. Therefore, the quality of this study and respective description was improved and the manuscript can be accepted in the present form.
Reviewer 3 Report
Comments and Suggestions for Authors
I am delighted to accept the proposed scientific work, as the authors have thoroughly and diligently addressed all revisions and changes suggested to them during the review process. The authors' responses have been highly constructive, and their willingness to incorporate requested suggestions and clarifications has significantly improved the quality and clarity of the manuscript. This commitment to scientific excellence and meticulous attention to detail reflect the professionalism and dedication of the authors to their work. I am convinced that the contribution of the article in question will be valuable to the scientific community.